# Extracellular Vesicles in Atherosclerosis: State of the Art

**DOI:** 10.3390/ijms25010388

**Published:** 2023-12-27

**Authors:** Wioletta Olejarz, Karol Sadowski, Klaudia Radoszkiewicz

**Affiliations:** 1Department of Biochemistry and Pharmacogenomics, Faculty of Pharmacy, Medical University of Warsaw, 02-091 Warsaw, Poland; s082684@student.wum.edu.pl; 2Centre for Preclinical Research, Medical University of Warsaw, 02-091 Warsaw, Poland; 3Translational Platform for Regenerative Medicine, Mossakowski Medical Research Institute, Polish Academy of Sciences, 02-106 Warsaw, Poland; kradoszkiewicz@imdik.pan.pl

**Keywords:** atherosclerosis, endothelial cells, extracellular vesicles, CVD, MSC, EPC

## Abstract

Atherosclerosis is a chronic inflammatory disease driven by lipid accumulation in the arteries, leading to narrowing and thrombosis that causes mortality. Emerging evidence has confirmed that atherosclerosis affects younger people and is involved in the majority of deaths worldwide. EVs are associated with critical steps in atherosclerosis, cholesterol metabolism, immune response, endothelial dysfunction, vascular inflammation, and remodeling. Endothelial cell-derived EVs can interact with platelets and monocytes, thereby influencing endothelial dysfunction, atherosclerotic plaque destabilization, and the formation of thrombus. EVs are potential diagnostic and prognostic biomarkers in atherosclerosis (AS) and cardiovascular disease (CVD). Importantly, EVs derived from stem/progenitor cells are essential mediators of cardiogenesis and cardioprotection and may be used in regenerative medicine and tissue engineering.

## 1. Introduction

Atherosclerosis is a chronic inflammatory disease driven by lipid accumulation in the arteries, leading to narrowing and thrombosis that causes mortality [1]. Emerging evidence has confirmed that AS affects younger people and is involved in the majority of deaths worldwide [2,3]. Vascular inflammation contributes to atherogenesis by activating molecular and cellular pathways and plays a key role in atherosclerosis plaques, which are unstable and can rupture, leading to stroke or myocardial infarction (MI) [4]. The molecular mechanisms associated with thrombotic complications of atherosclerosis have evolved far beyond the ‘vulnerable plaque’ concept [5]. This advance in knowledge has opened a path to efficient therapeutic interventions promising an improvement in the prevention and treatment of CVD [6]. However, the modulation of inflammatory components in atherosclerosis by gene therapy and targeting the effects on plaques remain as difficult challenges [7]. Extracellular vesicles (EVs) are lipid bilayer-enclosed structures secreted by endothelial cells, immune cells, and other cardiovascular tissues [8]. They are important mediators of intercellular and extracellular communications because they carry protein, DNA, mRNA, miRNA, non-coding RNA, lipids, and other bioactive molecules and deliver this genetic information to target cells [9,10]. miRNAs play a pivotal role in cardiogenesis, heart regeneration, and the pathophysiology of CVD [11]. miRNA transfer is important in cardiovascular systems and diseases because it modulates atherosclerosis [12]. The activation of cellular changes and the modulation of disease phenotypes are caused by EV miRNA transfer [13]. EVs can modulate phenotypes and cellular functions by regulating epigenetic alteration, transcription, and translation [14,15]. They have potential as biomarkers in diagnosis and prognosis as well as monitoring therapy in individuals with CVD and as new therapeutic targets and/or drug delivery vehicles [16,17,18].

### 1.1. Minimal Information for Studies of EVs (MISEV)

The International Society for Extracellular Vesicles (ISEV) has developed MISEV standardization in the EV research community [19,20]. In 2014, the MISEV2014 was provided, covering EV separation and isolation, characterization, and functional studies, and it was updated in 2018 (MISEV2018). The updated MISEV2018 offers vital suggestions, emphasizing precise terminology, characterization, and transparent reporting. In brief, MISEV18 describes “extracellular vesicle” as the preferred term and states that subtypes should be defined by their biochemical and physical characteristics and/or sources and conditions. Another conclusion is that diverse EV separation techniques require detailed reporting for replicability. Due to the development of EV characterization possibilities, protein and lipid markers seem to be crucial for demonstrating generic EV structure, adapted according to cellular origin and claim specificity. The authors also underline that demonstrating EV-associated functions demands rigorous evidence, emphasizing independence from cell–cell contact and differentiation from soluble factors. It is worth mentioning that the ISEV endorses the EV-TRACK knowledgebase, aligning with the MISEV for enhanced rigor [21,22].

### 1.2. Extracellular Vesicle (EV) Structure and Functions

EVs include exosomes, apoptotic bodies (ABs), and microvesicles (MVs), which play a crucial role in physiology and pathology [23]. They can be released by most cell types and detected in most body fluids, like plasma, urine, and saliva [24]. EVs have immunoregulatory functions in innate and adaptive immunity, including inflammation, antigen presentation, and the development and activation of B and T cells [25,26]. EVs have natural stability in circulation, biocompatibility, low toxicity, and low immunogenicity [27]. EVs contain tetraspanins (CD9, CD63, and CD81), membrane transport and fusion proteins (GTPases, annexins, and flotillin), heat shock proteins (Hsp60, Hsp70, and Hsp90), proteins involved in multivesicular body biogenesis (Alix andTSG101), cytoskeletal proteins (actin, cofilin, and tubulin), adhesion molecules (integrin-α and -β and P-selectin), glycoproteins (β-galactosidase, O-linked glycans, and N-linked glycans), growth factors and cytokines (TNF-α, TGF-β, TNF-related apoptosis-inducing ligand (TRAIL), and other signaling receptors (Fas ligand (FasL), TNF receptor, and transferrin receptor (TfR)) [28] (Figure 1).

## 2. EVs in Atherosclerosis and Cardiovascular Diseases

Atherosclerosis is an inflammatory process of the small- and medium-sized arteries and is primarily responsible for most CVD. AS is characterized by the accumulation of lipids due to impaired cholesterol metabolism, fibrous elements, and vascular calcification (VC) in the arteries. This process occurs due to inflammation, the oxidation of LDL particles, endothelial dysfunction, foam cell formation, proliferation, and the migration of vascular smooth muscle cells (VSMCs) [29]. The preceding vascular modifications are also affected by extracellular matrix molecules, soluble factors, and interactions between endothelial cells (EC) and EVs [30]. EVs are associated with critical steps in atherosclerosis, cholesterol metabolism, immune response endothelial dysfunction, and vascular inflammation and remodeling [31,32]. Recent studies have focused on EVs’ contents and biological functions, particularly their potential effects on atherosclerosis and the associated cholesterol metabolism [31].

Endothelial- and platelet-derived EVs have been shown to be independent biomarkers for coronary artery disease (CAD) [33,34,35,36]. Patients with endothelial dysfunction or atherosclerosis have demonstrated increased levels of EVs levels, which suggests that they are potential disease markers [37,38,39]. Moreover, it has also been proven that EC-derived EVs can affect the initial steps of atherosclerosis by aggravating endothelial barrier dysfunction, which leads to the transfer of Src kinase and the subsequent destabilization of tight junction proteins [36,40]. Platelet-derived EVs interact with subendothelial elements, so they participate in the phenotypic modulation of immune and vascular cells [41]. Regarding macrophages, studies have shown that platelet EVs can reduce macrophage reactivity by altering their differentiation into the M2 phenotype [42,43]. Additionally, these EVs hinder the production of nitric oxide (NO) in ECs, which worsens endothelial function [44]. Oxidized lipoproteins can modify the cargo of EVs produced by ECs. This alteration promotes the inclusion of proinflammatory substances, triggering the expression of adhesion molecules like intercellular adhesion molecule 1 (ICAM-1) and vascular cell adhesion molecule 1 (VCAM-1) [45,46,47,48] (Figure 2).

EVs can interact with platelets and monocytes, thereby influencing endothelial dysfunction, atherosclerotic plaque destabilization, and the formation of thrombous [49,50,51,52,53]. Plaque ruptures are mainly caused by fibrous cap weakening and the destabilization of atherosclerotic plaque caps by circulating EVs [54]. The breakdown of the extracellular matrix and the destabilization of plaques have been attributed to MMPs which are carried by EVs as their cargo [55,56]. EC-derived EVs have been shown to carry MMP-2, MMP-9, MMP-10, and MMP-14 on their external surface [57,58,59]. The cell source of EVs can affect their proteolytic cargo and fibrous cap integrity. For example, microvesicles isolated from atherosclerotic lesions express the metalloprotease TNF-α-converting enzyme (TACE/ADAM-17) [60,61]. Investigations have also detected the existence of EVs within calcified plaques and demonstrated their participation in the associated biological processes [62,63,64,65]. Endothelial EVs have been documented to exhibit elevated levels of calcium and bone morphogenetic protein 2 (BMP-2), leading to calcification and the induction of an osteogenic phenotype in smooth muscle cells (SMCs) [63].

Endothelial dysfunction, the initial step of atherogenesis, triggers the release of EVs, reflected in exosomal proteins and RNAs [66,67]. Notably, human atherosclerotic plaques have been observed to contain EVs originating from various cell types: leukocytes, macrophages, erythrocytes, lymphocytes, SMCs, and platelets [37,49,50,51,68,69]. Endothelial dysfunction, a stand-alone predictor of vascular disease, was evaluated in patients with stable CAD through quantitative measurements of CD31^+^/Annexin V^+^. It was shown that individuals with elevated levels of EVs developed significant cardiovascular or cerebral events afterward [70]. Endothelial cell-derived EVs may regulate the VSMC phenotype via cargos [71,72]. It was confirmed that miR-143/145-containing EVs derived from KLF2-expressing ECs reduced atherosclerotic lesions in ApoE^−/−^ mice [73]. By transferring miR10a, exosomes derived from ECs can modulate monocyte activation [74]. They can also reduce hypertension by regulating the VSMC phenotype via the ACE2/NF-κB/Ang II pathway [75]. Non-coding RNAs (ncRNAs), including microRNAs (miRNAs), long non-coding RNAs (lncRNAs), and circular RNAs (circRNAs), which are encapsulated in EVs, are involved in AS and VC [76,77]. ncRNAs shuttled via EVs may control post-hypoxia tissue survival and remodeling by regulating cell injury and inflammatory responses in MI and stroke [78,79]. Cardiomyocytes increase the secretion of EVs during stress, such as hypoxia, inflammation, or injury [80]. Cardiac-derived EVs are rich in microRNAs and are key to controlling cellular processes. Importantly, EVs secreted by cardiac progenitor cells (CPCs) and cardiosphere-derived cells are essential mediators of cardiogenesis and cardioprotection. Therefore, EVs from stem cells could exert a therapeutic effect on the damaged heart [81].

**Figure 2 ijms-25-00388-f002:**
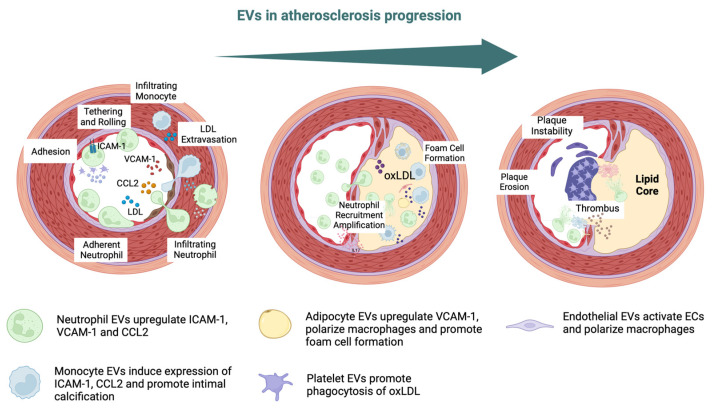
EVs in atherosclerosis progression. Intercellular adhesion molecule 1 (ICAM-1), vascular cell adhesion molecule-1 (VCAM-1), chemokine C–C motif chemokine ligand 2 (CCL2), low-density lipoprotein (LDL), oxidized LDL (oxLDL) [48,82].

## 3. EVs as Biomarkers and Therapeutic Targets in AS and CVD

EVs have potential as prognostic and diagnostic biomarkers of atherosclerosis initiation, progression, and complications [83]. EVs shield their molecular cargo from degradation, contributing to the progression of plaques and making them an excellent therapeutic target [84]. EVs’ presence in early and advanced plaques indicates their involvement in both the initial and later stages of plaque formation in humans, providing a chance to use them as progression markers [49,50,85]. Evaluating vascular well-being through EVs may offer valuable insights for preventing atherosclerosis in individuals who show no symptoms but have early signs of the disease [86,87]. For instance, EVs originating from ECs reflect endothelial impairment and indicate potential cardiovascular results [35,70,88] (Figure 3).

Increased levels of EVs are linked to risk factors like smoking, diabetes, and high cholesterol [89]. Exosomes are valuable prognostic and diagnostic biomarkers in CVD [90]. Elevated levels of specific categories of procoagulant EVs have been observed in individuals with acute coronary syndrome (ACS) compared with healthy individuals, and this phenomenon is linked to plaque rupture [34,91,92,93,94,95,96,97,98,99,100]. EV levels in the blood, urine, and saliva have been linked to clinical risk in patients with stable CVD [33]. Table 1 presents EVs as potential markers in cardiovascular diseases [34,35,70,94,100,101,102,103,104,105,106,107,108,109,110,111,112].

According to the revascularization success rate achieved in the culprit artery through PCI, there is an overall reduction in the levels of circulating EVs 72 h after a STEMI (ST elevation myocardial infarction). According to the successful rate of PCI revascularization in the culprit artery, there is a global decrease in circulating EVs 72 h after STEMI [86,92].

## 4. miRNA as a Biomarker in AS and CVD

miRNA is a class of small regulatory RNAs that regulate protein expression in recipient cells in a paracrine fashion. A promising biomarker for AS diagnosis, related to the occurrence of adverse cardiovascular events, is miR-18a-5p [114]. Also, serum miR-211-5p and miR-675-3p are new diagnostic biomarkers that are associated with the poor prognosis of AS [115]. miR-637 showed important predictive value for the occurrence of future cardiovascular events [116]. Circulating miRNAs, such as miR-17, miR-17-5p, miR-29b, miR-30, miR-92a, miR-126, miR-143, miR-145, miR-146a, miR-212, miR-218, miR-221, miR-222, and miR- 361-5p, are promising biomarkers for the clinical diagnosis of atherosclerosis [117]. Furthermore, miR-122-5p and miR-223-3p might be markers of plaque instability [118].

Importantly, exosomal miRNAs play an important role in the pathophysiology of CAD and acute myocardial infarction (AMI). miR-942-5p, miR-149-5p, and miR-32-5p may serve as potential diagnostic biomarkers for stable coronary artery disease (SCAD) [119]. miR-1915-3p, miR-4,507, and miR-3,656 were significantly less expressed in AMI samples than in SCAD [120]. Exosomal miR-152-5p and miR-3681-5p may serve as potential biomarkers for ST-segment elevation MI [121]. Also, MV miR-126 and miR-199a expression can predict the occurrence of cardiovascular (CV) events in patients with SCAD [110] Figure 4.

On the other hand, EV microRNAs are key players in cardiac regeneration and have cardioprotective and regenerative properties for cardiac and non-cardiac cells, stem cells, and progenitor cells [80,122]. It was found that exosomal microRNAs may deliver signals to mediate cardiac repair after MI [123]. It was shown that miR-22 derived from bone marrow mesenchymal stem cell-derived EVs (MSC-EVs) may reduce fibrosis and decrease apoptosis in mouse models of MI [124]. miR-19a and miR-221 from GATA-4 overexpressing MSCs also showed cardioprotective effects [125]. Cardiac progenitor cell-derived EVs (CPC-derived EVs) enriched in miR-210 and miR-132 may prevent stress-induced apoptosis in HL1 cardiomyocytes [126]. By restoring the miR-21/PDCD4 pathway, CPC-derived EVs inhibited oxidative stress-related apoptosis in H9c2 cells [127]. miR-210 and miR-132 play an important role in improving angiogenesis and cardiac function in MI models [128,129].

## 5. EVs in Tissue Regeneration and Repair in AS and CVD

Cell therapy holds great promise for treating diseases that are irreversible or incurable at the moment [130]. This is due to the fact that stem cells exhibit multilineage differentiation and self-renewal; moreover, they possess several immunomodulatory functions and promote tissue regeneration [131]. The success of regenerative medicine is based on two goals: one includes the direct restoration of the damaged tissue by cell transplantation; the other one is to repair the tissue via neurotrophic or immunomodulatory factors produced by cells or gap junction formation [132,133]. In many studies, the regenerative potential of stem cells has been seen in EV release [130,134].

It has been confirmed that EVs serve as efficient carriers of molecular cargo and are prime candidates for tissue engineering and regenerative medicine because of their favorable biological properties, including biocompatibility and stability [135,136]. EVs may modulate the activity of the immune system, playing an important role in regulating inflammation [25]. They exhibit a broad spectrum of immunomodulatory and pro-regenerative activity. By regulating cell proliferation, they are crucial for local tissue repair [137].

Via EV stimulation, many regenerative process-linked signaling pathways have been identified, including Wnt/β-catenin; PI3K/Akt; Notch; TGF-β/Smad; Hedgehog; CaMKII; and Efna3, AMPK and JAK2/STAT3/NF-κB [138,139,140,141] (Figure 5). For example, in MI, EVs from human umbilical cord MSCs were shown to stimulate angiogenesis through the microRNA-423-5p/EFNA3 pathway [142]. Other findings have suggested that EVs derived from human bone marrow MSCs effectively enhance cutaneous wound healing by inhibiting the TGF-β/Smad signaling pathway. Notably, this study revealed that the EVs induced a significant downregulation of TGF-β1, Smad2, Smad3, and Smad4 expression while concurrently upregulating TGF-β3 and Smad7 expression in the TGF-β/Smad signaling pathway [143]. Moreover, the pivotal role of the Wnt/β-catenin pathway as a key regulator of proangiogenic effects in EVs derived from tubular epithelial cells (TEC) was determined [141]. To develop an effective strategy for promoting vascularized bone regeneration, the impact on angiogenesis was also studied in EVs from hypoxia-induced MSCs. These EVs were observed to significantly enhance the proliferation, migration, and angiogenesis of human umbilical vein endothelial cells (HUVECs). The study revealed that HIF-1α induced the upregulation of miR-210-3p, which subsequently inhibited EFNA3 expression, leading to an enhancement in phosphorylation in the PI3K/AKT pathway [144]. Recent reports highlighted human placenta-derived MSC EVs as a promising therapeutic approach for nerve injury-induced neuropathic pain. The study reported that miR-26a-5p in EVs plays a regulatory role in the Wnt5a/Ryk/CaMKII/NFAT pathway. This regulation partially contributes to analgesic effects through anti-neuroinflammation [145]. Another study showed that EVs derived from MSCs effectively reduced apoptosis and alleviated tissue damage induced by middle cerebral artery occlusion (MCAO) in rats. This protective mechanism was shown to be likely mediated through the regulation of the AMPK pathway and the JAK2/STAT3/NF-κB signaling pathways. The findings highlight the neuroprotective potential of EVs in the context of MCAO, suggesting a promising therapeutic opportunity for ischemic stroke treatment [140] (Figure 5).

EV phenotype can vary depending on the stem cell source. For example, EVs derived from endothelial progenitor cells (EPCs) may be used in regenerative medicine, especially in cardiac repair, because of their rich anti-inflammatory properties, while the EVs isolated from cardiomyocytes or cardiosphere-derived cells were proposed to be applied mainly due to their pro-proliferative effects [146]

EPCs seem to promote wound healing via angiogenesis and collagen synthesis stimulation [147]. EPCs are characterized by strong growth potential and the ability to differentiate into endothelial cells [148,149]. It was shown that EPC-derived exosomes presented a cardioprotective effect in angiotensin II-induced hypertrophy and apoptosis [150]. Moreover, they can control angiogenesis by promoting endothelial cell survival, proliferation, and organization [151].

The enhancement of angiogenesis was also reported to be a property of cardiomyocyte EVs after oxidative and metabolic stress [11]. Under physiological conditions, the released EVs regulated cell survival and growth. Other candidates for myocardial disease treatment, cardiac progenitor cells (CPCs), seem to be very attractive because of their origin and the fact that they contribute to maintaining the population of cardiac myocytes and coronary vessels under normal conditions [152,153]. Their protective effect can be achieved both by their further differentiation and paracrine abilities [154]. CPCs, however, can hamper myocardial regeneration due to the limited proliferative potential [155]. By contrast, exosomes derived from cardiospheres (CDCs) stimulate cardiomyocytes’ proliferative, angiogenic, and anti-apoptotic potential. CDCs are clusters composed of cardiomyocytes, cardiac stem cells, CPCs, endothelial cells, and smooth muscle cells [156]. It was observed recently that CDCs’ EVs seem to reduce the inflammatory response by suppressing monocytes’ expression of CCR2 in patients with acute MI, offering a new prospect for existing clinical therapies for elevated pro-inflammatory responses following this incident [157].

The beneficial therapeutic impacts of MSCs are largely associated with their exosomal paracrine functions [158]. Mesenchymal stem cells (MSCs) are multipotent stem cells with self-renewal abilities, and they induce tissue regeneration by their direct recruitment into injured tissues, including the heart [159,160]. MSCs may be found in many tissue sources, including the bone marrow, adipose tissue (AD-MSCs) and umbilical cord blood, Wharton’s jelly (WJ-MSCs), and even the brain, placenta, liver, kidney, lungs, and thymus [161]. Due to the great access to many MSC sources, these cells have attracted the attention of researchers and are commonly used in experiments. Recently, it was shown that MSCs can release several EVs whose activity in regenerative processes is similar to that of the MSCs [162]. Moreover, it was confirmed that MSC-EVs are cardioprotective [163]. EVs from these cells are beneficial in various ways—for example, their anti-inflammatory, anti-apoptotic, and anti-remodeling actions have regenerative and neovascular properties [164]. MSC-EVs were also reported to decrease oxidative stress [165]. MSCs, through their advantageous properties, such as accelerating wound closure, improving re-epithelialization, elevating angiogenesis, suppressing inflammation, and modulating extracellular matrix (ECM) remodeling, can accelerate wound repair [166]. MSC-derived EVs can shuttle bioactive molecules, causing the exchange of genetic information and reprogramming of the recipient cells. Importantly, EVs could mediate and induce cell differentiation/self-renewal [167].

The most attractive MSCs regarding cardiovascular diseases seem to be adipose tissue-derived mesenchymal stem cells (AD-MSCs). AD-MSCs are relatively easy to isolate and culture and are characterized by a high proliferation rate and multilineage differentiation [168,169]. Despite their relatively low survival rate in a cardiac niche, their paracrine potential is known to be the most promising one nowadays [146,168,170]. Luo et al. reported that AD-MSC-derived exosomes can protect myocardial cells from apoptosis, inflammation, and fibrosis, and they can promote angiogenesis, thereby preventing myocardial damage [171].

MSC-derived EVs may be used as an alternative MSC-based therapy in regenerative medicine and tissue engineering [172,173]. They pave the way to an effective, cost-efficient, and safe therapeutic option in cell-free regenerative medicine, presumably offering a viable alternative to MSCs. Therefore, understanding their functional mechanisms is essential to fully address the regenerative potential of MSC-derived EVs in therapeutic applications.

## 6. Conclusions and Future Directions

It has been confirmed that EVs are implicated in key aspects of atherosclerosis progression by regulating pathophysiologic pathways like inflammation, angiogenesis, and senescence [174,175]. They play an important role in physiological and pathological conditions [176]. EVs have potential as biomarkers in the diagnosis, prognosis, and monitoring of atherosclerosis in patients and as new therapeutic targets and/or drug delivery vehicles [17]. It is worth mentioning that altered levels of EVs have been evidenced in patients suffering from atrial fibrillation [177]. They may be clinically useful as therapeutic targets in cardiovascular diseases. A novel therapeutic approach for atherosclerosis prevention and treatment is the precise targeting of genes involved in lipoprotein metabolism. Therefore, the roles of EVs-derived RNAs as therapeutic products or theranostic markers demonstrate unequivocally encouraging benefits. The homeostatic functions of EVs may facilitate the development of new regenerative therapies. Importantly, EVs obtained from various stem/progenitor cells are safer and more effective in repairing damaged tissues [178]. They play an important role in intracardiac communication and present huge potential in regenerative medicine [179]. EPCs represent more biocompatible, less immunogenic populations and have potential as a cell-free therapy for CVD [180]. Also, tissue engineering and targeted therapies by using MSCs are capable of reducing inflammation and increasing regenerative potential. The MISEV has established essential principles while fostering flexibility, collaboration, and continuous improvement in EV research. By collectively advancing isolation, detection, and standardization, the full potential of EVs can be unlocked, enhancing our understanding of their roles in health and disease. However, it should be emphasized that despite the success in engineering EVs toward clinical applications, the use of EVs is still hampered by challenges encountered in EV isolation and detection and the lack of standardization methods.

## Figures and Tables

**Figure 1 ijms-25-00388-f001:**
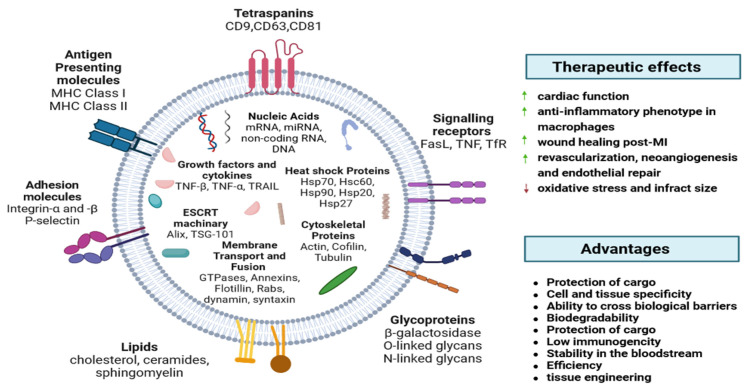
EVs’ exterior and interior components. Major histocompatibility complex (MHC), heat shock proteins (Hsp), tumor susceptibility gene (TSG), tumor necrosis factor (TNF), TNF-related apoptosis-inducing ligand (TRAIL), Fas ligand (FasL), transferrin receptor (TfR) [28].

**Figure 3 ijms-25-00388-f003:**
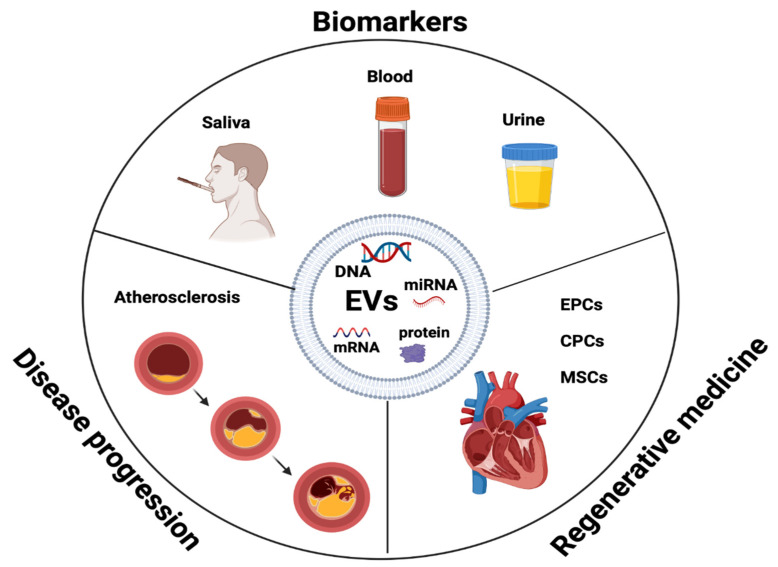
EVs as biomarkers in disease progression and regenerative medicine. Endothelial progenitor cells (EPCs), cardiac progenitor cells (CPCs), mesenchymal stem cells (MSCs).

**Figure 4 ijms-25-00388-f004:**
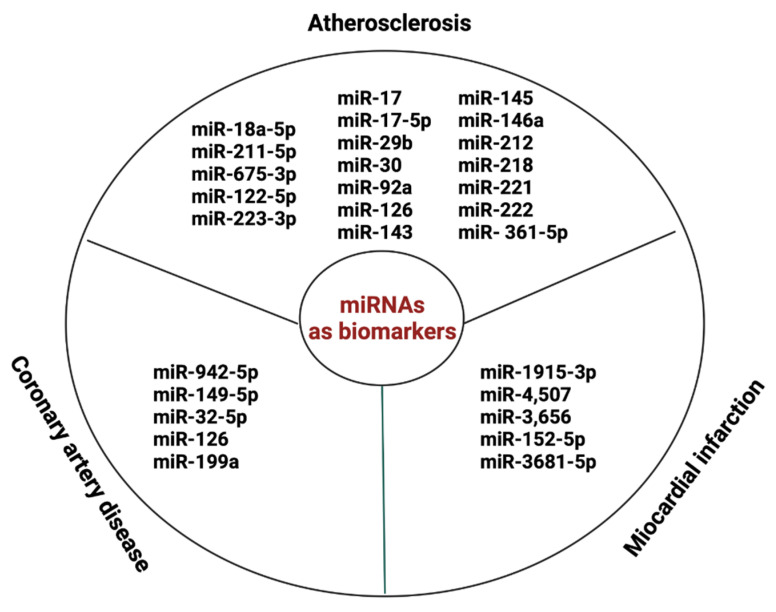
miRNAs as biomarkers in AS and CVD.

**Figure 5 ijms-25-00388-f005:**
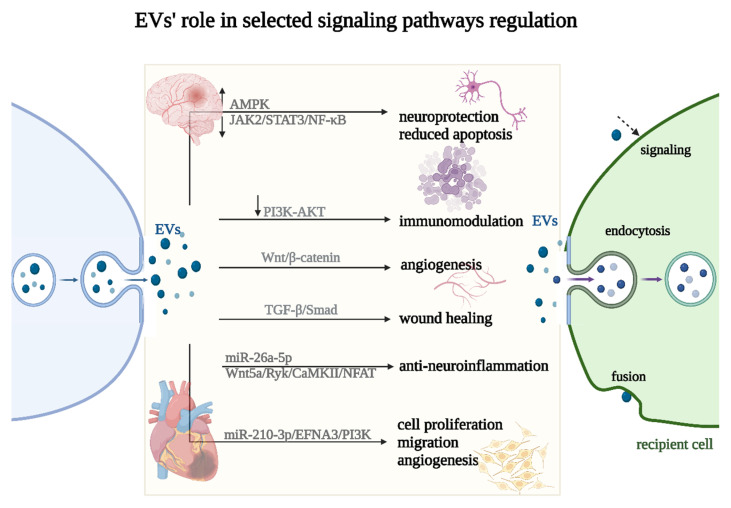
Selected signaling pathways’ regulation (↑—upregulation, ↓—downregulation) by EVs. EVs can impact the recipient cell phenotype using three different uptake mechanisms: endocytosis, fusion, or signaling (marked with a dashed arrow).

**Table 1 ijms-25-00388-t001:** EVs as markers for diagnosing and predicting the progression of atherosclerosis.

Disease	EV Subtype	Reference
Coronary artery disease (CAD)	CD31^+^/Annexin V^+^ EV	[70]
	eEV (CD31^+^), pEV (CD42b^+^)	[110]
	eEV (CD31^+^/Annexin V^+^)	[35]
	eEV (CD31^+^ and CD51^+^)	[34]
	eEV (CD31^+^)	[109]
	eEV (CD31^+^/41^−^, CD62E^+^, CD144^+^)	[108]
	eEV (CD62E^+^, CD31^+^)	[107]
Coronary heart disease	eEV (CD31^+^/CD42^−^, CD144^+^)	[106]
	pEV (CD41^+^), eEV (CD144^+^)	[105]
	eEV (CD31^+^ and CD146^+^)	[94]
Stable angina	eEV (CD31^+^), pEV (CD41^+^)	[104]
Acute coronary syndrome	pEV (CD31^+^, CD41a^+^)	[103]
	eEV (CD146^+^)	[102]
	eEV (CD144^+^), ErEV (CD235a^+^), pEV (CD41a^+^)	[101]
Endothelial dysfunction	eEV (CD31^+^/CD42b^−^)	[100]
	eEV (CD31^+^/AnnV^+^)	[35]
Subclinical atherosclerosis	lEV (CD3^+^/CD45^+^)	[112]
	pEV (TSP1^+^/CD142^+^)	[111]

Cluster of differentiation (CD); endothelial-derived EV (eEV); erythrocyte-derived EV (ErEV); leucocyte-derived EV (IEV); platelet-derived EV (pEV); thrombospondin 1 (TSP1) [113].

## Data Availability

Not applicable.

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
