# Peer review of "Extracellular Vesicles in Atherosclerosis: State of the Art"

_ijms, 2023, doi:10.3390/ijms25010388_

Round 1

Reviewer 1 Report

Comments and Suggestions for Authors

In this review, Dr Olejarz et al try to provide an update on EVs in atherosclerosis by summarizing studies reporting EVs as biomarker source for diagnosis/prognosis, therapeutic target and therapeutics for regenerative medicine. While the topic is interesting, the manuscript is not well organized and the novelty of current review manuscript is unclear. The authors are advised to revise the manuscript extensively according to comments as follow:

Major

1.    The Abstract is not well structured. The last 2 sentences stating regenerative medicine and MSC-EVs for cardiac repair/regeneration do not sound logically adhesive to the rest of the Abstract, instead, very abrupt and disconnected. 

2.    In the Introduction, a brief summary of the review background, novelty, and structure should be provided. In particular, in terms of novelty, the authors should discuss the new advances/angle of current manuscript compared to other recent similar reviews such as the extensive review article entitled “Extracellular vesicles in cardiovascular disease” by Huang et al (PMID: 34229852), so what’s new? The authors may also refer to those existing reviews for some contents in the current manuscript to avoid repetition. Instead, the authors can focus on new progress/update in the field which is unfortunately not apparent in the current version. 

3.    Figure 1, it could be more effective to provide a compilation of all the figure legends that showcase the exterior and interior components. Categorizing the exterior and interior components as proteins, lipids, ligands, and genes can improve clarity. Additionally, it would be helpful to indicate the functions of those EV components or “advantages”. To express the up-regulate or down-regulate therapeutic effects, using distinct colors of arrows can be useful.

4.    Figure 1 is cited in a section discussing a very general aspect of EVs, i.e. structure and function, the figure content obviously does not fit in this section, in other word, the figure was not properly discussed in the text. 

5.    In the section on EVs in atherosclerosis and cardiovascular diseases (CVD), see my comments above regarding existing similar review articles. In addition, beside the endothelial- and platelet-derived EVs, the authors should provide an overview of different types of EVs as well as their respective sources and related functions in the progression of atherosclerosis, as indicated in Figures 2 & 3. Also, the text describing figure 2 that presents different types of EVs should be rephrased according to the order/sequence.

6.    It is suggested to illustrate the EVs in the atherosclerosis progression more comprehensively in Figure 2. Since the progression of atherosclerosis involves several steps, ranging from foam cell formation to plaque rupture, it would be more effective to present the individual stages of atherosclerosis, along with the corresponding functions of EVs, to clearly demonstrate the influence of EVs under different conditions. Meanwhile, Figure 2 requires figure legends to illustrate the icons such as macrophages, endothelial cells, or plaques, etc. Try to avoid much text in the Figure.

7.    In Figure 3, the biomarkers section should include icons for all sample types, not just blood.

8.    In Figure 3, please provide separate illustrations for the initiation, promotion, and progression stages of atherosclerosis using appropriate icons to represent the progression stages. Also, it seems that the subtitle "Disease progression" and the text description "progression" in the figure are redundant. Please clarify to avoid confusion.

9.    In the "miRNA as biomarkers in AS and CVD" section, it would be helpful to include a table summarizing related miRNAs and their associated pathophysiology.

10. In the section on "EVs in tissue regeneration and repair in AS and CVD," the following pathways are mentioned: Wnt/β-catenin, PI3K/Akt, Notch, TGF-β/Smad, Hedgehog, CaMKII, Efna3, 221 AMPK, and JAK2/STAT3/NF-κB. Please create pathway maps and explain how EVs regulate the expression of key proteins.

11. In the conclusion and future direction section, potential methods to handle isolation, detection, and standardization should be provided to guide further research.

Minor

12. Too many jargons make the manuscript difficult to read. The authors shall ensure all abbreviations given full names at their first appearance. 

Comments on the Quality of English Language

English writing is ok but too many jargons make the manuscript difficult to read. The authors shall ensure all abbreviations given full names at their first appearance. 

Author Response

Comments and Suggestions for Authors

In this review, Dr Olejarz et al try to provide an update on EVs in atherosclerosis by summarizing studies reporting EVs as biomarker source for diagnosis/prognosis, therapeutic target and therapeutics for regenerative medicine. While the topic is interesting, the manuscript is not well organized and the novelty of current review manuscript is unclear. The authors are advised to revise the manuscript extensively according to comments as follow:

Response: Thank you very much for review our manuscript, positive opinion and important comments. We have corrected stylistic imperfections through the manuscript. We have added new information about EVs in atherosclerosis and cardiovascular diseases.

Our response in manuscript is shown in red text.

Major

  1. The Abstract is not well structured. The last 2 sentences stating regenerative medicine and MSC-EVs for cardiac repair/regeneration do not sound logically adhesive to the rest of the Abstract, instead, very abrupt and disconnected.

Response: We have corrected the Abstract

  1. In the Introduction, a brief summary of the review background, novelty, and structure should be provided. In particular, in terms of novelty, the authors should discuss the new advances/angle of current manuscript compared to other recent similar reviews such as the extensive review article entitled “Extracellular vesicles in cardiovascular disease” by Huang et al (PMID: 34229852), so what’s new? The authors may also refer to those existing reviews for some contents in the current manuscript to avoid repetition. Instead, the authors can focus on new progress/update in the field which is unfortunately not apparent in the current version. 

Response: We have added more information about EVs by Huang and other authors in the Introduction

  1. Figure 1, it could be more effective to provide a compilation of all the figure legends that showcase the exterior and interior components. Categorizing the exterior and interior components as proteins, lipids, ligands, and genes can improve clarity. Additionally, it would be helpful to indicate the functions of those EV components or “advantages”. To express the up-regulate or down-regulate therapeutic effects, using distinct colors of arrows can be useful.

Response: We have corrected Figure 1.

  1. Figure 1 is cited in a section discussing a very general aspect of EVs, i.e. structure and function, the figure content obviously does not fit in this section, in other word, the figure was not properly discussed in the text. 

Response: We have added MISEV.

  1. In the section on EVs in atherosclerosis and cardiovascular diseases (CVD), see my comments above regarding existing similar review articles. In addition, beside the endothelial- and platelet-derived EVs, the authors should provide an overview of different types of EVs as well as their respective sources and related functions in the progression of atherosclerosis, as indicated in Figures 2 & 3. Also, the text describing figure 2 that presents different types of EVs should be rephrased according to the order/sequence.

Response: We have added more information  according to the comments.

  1. It is suggested to illustrate the EVs in the atherosclerosis progression more comprehensively in Figure 2. Since the progression of atherosclerosis involves several steps, ranging from foam cell formation to plaque rupture, it would be more effective to present the individual stages of atherosclerosis, along with the corresponding functions of EVs, to clearly demonstrate the influence of EVs under different conditions. Meanwhile, Figure 2 requires figure legends to illustrate the icons such as macrophages, endothelial cells, or plaques, etc. Try to avoid much text in the Figure.

Response: We have corrected Figure 2 according to the comments.

  1. In Figure 3, the biomarkers section should include icons for all sample types, not just blood.

Response: We have corrected Figure 3 according to the comments.

  1. In Figure 3, please provide separate illustrations for the initiation, promotion, and progression stages of atherosclerosis using appropriate icons to represent the progression stages. Also, it seems that the subtitle "Disease progression" and the text description "progression" in the figure are redundant. Please clarify to avoid confusion.

Response: We have corrected Figure 3

  1. In the "miRNA as biomarkers in AS and CVD" section, it would be helpful to include a table summarizing related miRNAs and their associated pathophysiology.

Response: We have summarized information about miRNAs in AS and CVD on Figure 4 according to the comments.

  1. In the section on "EVs in tissue regeneration and repair in AS and CVD," the following pathways are mentioned: Wnt/β-catenin, PI3K/Akt, Notch, TGF-β/Smad, Hedgehog, CaMKII, Efna3, 221 AMPK, and JAK2/STAT3/NF-κB. Please create pathway maps and explain how EVs regulate the expression of key proteins.

Response: We have created figure 5 according to the comment.

  1. In the conclusion and future direction section, potential methods to handle isolation, detection, and standardization should be provided to guide further research.

Minor

  1. Too many jargons make the manuscript difficult to read. The authors shall ensure all abbreviations given full names at their first appearance. 

Response: We have corrected ms according to the comment.

Reviewer 2 Report

Comments and Suggestions for Authors

This review summarizes a list of 175 publications on extracellular vesicles in relation to atherosclerosis. It is not really clear how this review can be titled “state of the art”. Actually, the manuscript is poorly written and structured. Too many citations with not enough details and explanation of the importance of the findings. Consequently, it is a frustrating manuscript to read. There is some good intention, but it needs a major rewriting in order to achieve its goal. Just one minor detail as an example, at line 79, the authors mentioned “ Several studies indicate a favorable connection between the number of EVs and cardiovascular condition…”, however, most EV mentioned in the same paragraph are either deleterious or directly associated with worst CV condition.

Comments on the Quality of English Language

Poor. Needs extensive revision.

Author Response

Comments and Suggestions for Authors

This review summarizes a list of 175 publications on extracellular vesicles in relation to atherosclerosis. It is not really clear how this review can be titled “state of the art”. Actually, the manuscript is poorly written and structured. Too many citations with not enough details and explanation of the importance of the findings. Consequently, it is a frustrating manuscript to read. There is some good intention, but it needs a major rewriting in order to achieve its goal. Just one minor detail as an example, at line 79, the authors mentioned “ Several studies indicate a favorable connection between the number of EVs and cardiovascular condition…”, however, most EV mentioned in the same paragraph are either deleterious or directly associated with worst CV condition.

Response: Thank you very much for review our manuscript and important comments. We have corrected imperfections and errors through the manuscript. We have added new information about EVs in atherosclerosis and cardiovascular diseases (shown in red text).

Round 2

Reviewer 1 Report

Comments and Suggestions for Authors

All my concerns have been well addressed.